# Detection and Reconstruction of Poor-Quality Channels in High-Density EMG Array Measurements

**DOI:** 10.3390/s23104759

**Published:** 2023-05-15

**Authors:** Emma Farago, Adrian D. C. Chan

**Affiliations:** Department of Systems and Computer Engineering, Carleton University, Ottawa, ON K1S 5B6, Canada; adrian.chan@carleton.ca

**Keywords:** HD-EMG, signal quality analysis, electromyography, interpolation, outlier detection

## Abstract

High-density electromyography (HD-EMG) arrays allow for the study of muscle activity in both time and space by recording electrical potentials produced by muscle contractions. HD-EMG array measurements are susceptible to noise and artifacts and frequently contain some poor-quality channels. This paper proposes an interpolation-based method for the detection and reconstruction of poor-quality channels in HD-EMG arrays. The proposed detection method identified artificially contaminated channels of HD-EMG for signal-to-noise ratio (SNR) levels 0 dB and lower with ≥99.9% precision and ≥97.6% recall. The interpolation-based detection method had the best overall performance compared with two other rule-based methods that used the root mean square (RMS) and normalized mutual information (NMI) to detect poor-quality channels in HD-EMG data. Unlike other detection methods, the interpolation-based method evaluated channel quality in a localized context in the HD-EMG array. For a single poor-quality channel with an SNR of 0 dB, the F1 scores for the interpolation-based, RMS, and NMI methods were 99.1%, 39.7%, and 75.9%, respectively. The interpolation-based method was also the most effective detection method for identifying poor channels in samples of real HD-EMG data. F1 scores for the detection of poor-quality channels in real data for the interpolation-based, RMS, and NMI methods were 96.4%, 64.5%, and 50.0%, respectively. Following the detection of poor-quality channels, 2D spline interpolation was used to successfully reconstruct these channels. Reconstruction of known target channels had a percent residual difference (PRD) of 15.5 ± 12.1%. The proposed interpolation-based method is an effective approach for the detection and reconstruction of poor-quality channels in HD-EMG.

## 1. Introduction

High-density electromyography (HD-EMG) enables the measurement of the electrical activity of muscles over a spatial distribution. HD-EMG signals are collected via arrays consisting of numerous (10–100) electrodes. HD-EMG has been applied to numerous fields involving muscle contraction, including kinematics [1], rehabilitation [2,3], diagnostics [2], user authentication [4], and control of prosthetic devices [4].

EMG signals are susceptible to contaminants (e.g., motion artifacts, power-line noise, electrocardiograms, random noise, and background spikes) and measurement errors (e.g., electrode lift) [5,6,7]. Since HD-EMG arrays contain a large number of electrodes, it can be difficult to completely avoid the presence of poor-quality channels when collecting HD-EMG data. HD-EMG arrays pose additional challenges, as the numerous closely-spaced electrodes have an increased risk of shorted connections with nearby electrodes due to misplaced electrode gel [8].

HD-EMG is still somewhat cumbersome and time-consuming to collect. Since HD-EMG may consist of hundreds of channels, poor measurements are often only noticed after data collection, and recollecting the data might not be possible if the participant has already departed [7,9]. Several approaches [8,10,11,12] have been developed to identify poor-quality channels in HD-EMG arrays. These methods are typically based on outlier detection, whereby poor-quality channels are distinguished as having markedly different features from the majority of the channels in the HD-EMG array.

Marateb et al. [8] developed outlier detection methods to identify poor-quality channels by training a local distance-based outlier factor (LDOF) model (a *k*-nearest neighbours-based classifier). Three input features were used: the cross correlation between each EMG channel and neighbouring channels and two spectral features designed to detect power-line and low-frequency noise. The model was evaluated using an upper-limb HD-EMG dataset with channels labelled as either “good” or “bad” by three expert raters. A total of 20 array sets were randomly selected for human rating from a dataset of 432 total array sets (12 participants × 3 force levels × 3 array locations × 4 muscle contraction types (i.e., flexion, extension, pronation, and supination)). Among the 20 signal sets, 100 out of 2400 channels were identified as “bad” by human raters. The proposed method had a reported sensitivity (recall) of 96.9% and a specificity of 96.4%.

Rojas-Martínez et al. [11] developed a threshold-based method of identifying poor-quality channels based on three features: (1) relative power of low-frequency contaminants (≤12 Hz), (2) relative power of power-line noise (50 Hz Europe/60 Hz North America and harmonics), and (3) the root mean square (RMS) of the signal. The algorithm was applied to the same dataset in [8] and had a reported sensitivity of 97.94% and a specificity of 99.46%.

Bingham et al. [12] identified channels contaminated with noise based on the normalized mutual information (NMI) between each EMG channel and every other channel in the HD-EMG array. Channels with lower NMI values compared to other channels in the array were identified as likely outliers. An HD-EMG dataset was recorded from the tibialis anterior muscle using two 4 × 8 electrode arrays. A total of 12 array sets (3 participants × 2 force levels × 2 arrays) were used to provide clean EMG data. Contaminated channels were simulated by applying white Gaussian noise to a clean HD-EMG array. This allowed for outlier detection for multiple signal-to-noise ratios (SNRs) and multiple numbers and distributions of outlier channels, as well as the ability to use the original uncontaminated EMG channel as a ground truth. The authors reported that noisy channels with SNRs of 0, 5, and 10 dB were reliably identified for groups of two, four, and eight noisy channels. Sensitivities and specificities were reported as 100% for groups of two, four, and eight noisy channels with SNRs of 0 and 5 dB. At 10 dB, sensitivities were 99.74%, 95.31%, and 71.61% for two, four, and eight noisy channels, respectively.

Following outlier detection, poor-quality channels could simply be discarded; however, downstream data processing often assumes a complete HD-EMG dataset, and recollecting the data is often undesirable. Reconstruction of poor or missing channels could assist in preserving the original dimensions of the HD-EMG array and simplify EMG clinical analysis or research [11,13]. Reconstruction of HD-EMG channels has been proposed based on interpolation methods [8,11,13,14], but there has been limited work on systematically investigating these methods.

Rojas-Martínez et al. [15] proposed triangle-based cubic interpolation to reconstruct RMS feature values for poor-quality HD-EMG channels and cubic spline interpolation [11] to develop activation maps that can be compared across participants for the RMS feature; however, this method was not validated (i.e., the accuracy of the reconstruction was not assessed). Afsharipour et al. [14] reconstructed poor-quality HD-EMG channels using “the interpolation of the eight neighboring channel within one inter-electrode distance” but did not provide further explanation as to how the interpolation was conducted, nor did they validate the method.

Farago et al. [13] evaluated interpolation techniques by comparing the percent residual difference (PRD) and correlation between a target channel (simulated as a missing channel to be reconstructed) and an interpolation of nearby electrodes. They determined that a two-dimensional (2D) spline interpolation using the nearest 24 electrodes to the missing channel provided the best interpolation result (median PRD = 12.0%, median correlation = 0.99). The interpolations of poor-quality channels were also found to have higher PRDs compared to other channels, suggesting that interpolation can be used as an approach for the detection and reconstruction of poor-quality channels.

In this paper, we developed a new process to automatically detect and reconstruct poor-quality HD-EMG channels via interpolation. Previous methods to detect poor-quality channels identified them as outliers within the entire array [8,11,12]. Our novel interpolated-based method, on the other hand, detects poor-quality channels in a local context, identifying them as outliers relative to their neighbouring channels. In addition, we paired our method for poor-quality channel detection with a method for reconstruction of poor-quality channels. This is the first time a complete process is presented and evaluated to detect and recover poor-quality channels in HD-EMG array recordings.

Figure 1 provides a block diagram for the process of detecting and reconstructing poor-quality channels. In Figure 1a, HD-EMG signals are acquired, with a poor-quality channel indicated in yellow. In Figure 1b, this poor-quality channel is detected using the proposed interpolation-based method. This poor-quality channel is then reconstructed via interpolation based on its neighbouring channels as illustrated in Figure 1c. Figure 1d shows the final HD-EMG array with the poor-quality channel reconstructed.

The process is compared with two other outlier detection methods for identifying poor-quality channels and evaluated with simulated and real HD-EMG data.

The remainder of this paper is organized as follows. In Section 2, the materials and methods are detailed. In Section 3, we evaluate and analyze the results. Section 4 provides a discussion, and Section 5 presents the conclusions.

## 2. Materials and Methods

### 2.1. EMG Database

HD-EMG isometric upper-limb recordings of approximately 2 min were obtained from a publicly available database [16]. The dataset consists of monopolar EMG recordings acquired with an OT Bioelettronica (Torino, Italy) EMG-USB 128-channel system with a sampling frequency of 2048 Hz filtered with a 3 dB bandwidth of 10–175 Hz. Electrode arrays of 8 × 15 electrodes were placed on the biceps and triceps (silver-plated, gel-filled circular electrodes; diameter = 5 mm; separation = 10 mm). HD-EMG recordings from 12 subjects performing isometric bicep and tricep contractions at 30% maximal voluntary contraction (MVC) were used to provide a total of 24 HD-EMG arrays. All participants were healthy male volunteers without a history of neuromuscular disorders, pain, or regular training of the upper limbs [16].

### 2.2. Poor-Quality Channel Ground Truth

The two authors (each with >5 years EMG experience) independently rated the signal quality of each EMG channel, providing a ground truth for the detection of poor-quality channels. Raters were presented with each EMG channel and the neighbouring eight channels in the time and frequency domains. Raters were also provided with a histogram of the mean absolute value (MAV) feature across the HD-EMG array, with the MAV of the EMG channel under evaluation marked on the histogram. As rating required the eight neighbouring channels, channels along the array borders were excluded (i.e., only the interior 6 × 13 electrode array was used). Raters were instructed to rate the channels into one of four categories: (0) poor quality: EMG has insufficient quality for use due to major contamination and/or low amplitude; (1) adequate quality: EMG has sufficient quality for use but has noticeable contamination and/or low amplitude; (2) good quality: EMG recommended for use, with minor contamination and/or moderate-to-high amplitude; or (3) excellent quality: EMG strongly recommended for use, with little to no contamination and high amplitude.

Each EMG channel was rated three times by each rater to allow for assessment of intrarater and inter-rater variability. The order of channel presentation was randomized, and the channels were assessed by each rater independently. Channels with an overall mean rating ≤0.5 were considered to be poor-quality channels. Of the 1872 channels studied (24 arrays with 6 × 13 channels), 19 were identified as poor-quality by human raters. Of the 24 arrays, 15 arrays contained 1 poor-quality channel, 2 arrays contained 2 poor-quality channels, and 7 arrays had no poor-quality channels.

### 2.3. Simulated Poor-Quality Channels

Simulated noisy channels were generated using a methodology based on the simulation described by Bingham et al. [12]. Noisy channels were generated by applying additive white Gaussian noise (WGN) to the HD-EMG channels that were rated as adequate or better (overall mean rating > 0.5). The 19 EMG channels that were identified by human raters as poor-quality were excluded from the simulation study. WGN was added to the EMG signals at SNR values ranging from −20 dB to 15 dB in increments of 5 dB. Various numbers of poor-quality channels were investigated: one, two, four, and eight. As in [12] two, four, and eight noisy channels were investigated as both randomly dispersed or present in a clump (contiguous channels). For each condition, noisy channels were created in 30 different locations for each array. This resulted in 30 locations × 12 subjects × 8 SNR levels × 7 noisy channel configurations = 20,160 samples.

### 2.4. Detection of Poor-Quality EMG Channels

Three poor-quality EMG channel detection methods were explored in this study: (1) the proposed interpolation-based method, (2) a detection method based on power features and the RMS feature described by Rojas-Martínez et al. [11], and (3) the NMI-based method described by Bingham et al. [12].

#### 2.4.1. Interpolation-Based Detection Method

The interpolation-based detection method is a rule-based method that uses the PRD of the interpolated signals. The PRD feature provides a quantitative measure of the difference between the recorded signal and the interpolated signal. A higher PRD feature indicates a greater difference between the two signals. For this method, the target channel is the channel under investigation and is the channel for which the interpolations are made. A nearest-neighbour interpolation was used, which simply selects an adjacent channel to interpolate the target channel. Four interpolations were produced using the four adjacent channels above, below, to the left, and to the right of the target channel. The PRD was calculated as follows:(1)$$PRD_{dir}\left(n\right)=\sqrt{\frac{\sum_{i=1}^{N}{\left(x\left(i\right)-y_{dir}\left(i\right)\right)}^{2})}{\sum_{i=1}^{N}{y_{dir}\left(i\right)}^{2}}}\times 100$$
where *n* is the channel number, x(i) is the recorded EMG signal of the target channel, $y_{dir}\left(i\right)$ is the interpolated EMG signal, and dir is the direction of the channel used for interpolation (above, below, left, or right). The four resulting PRDs were combined to create a composite feature, $PRD_{c}$, which is defined as follows:(2)$$\begin{array}{c} PRD_{c}\left(n\right)=min\{PRD_{above}\left(n\right),PRD_{below}\left(n\right),\\ PRD_{left}\left(n\right),PRD_{right}\left(n\right)\} \end{array}$$

A threshold for detecting poor-quality channels was determined as follows:(3)$$\begin{array}{c} th_{PRD}=min\{median\left(PRD_{c}\right)+\tau ,\\ median\left(PRD_{c}\right)+\Phi std\left(PRD_{c}\right)\} \end{array}$$
where τ and Φ are tunable constants. A poor-quality channel does not correspond well to its interpolations; that is, it has a high $PRD_{c}$, which is detected as being at least Φ standard deviations greater than the median $PRD_{c}$ across all channels in the array. There are cases in which the standard deviation is small (e.g., no poor-quality channels present), so a fixed minimum threshold set by τ is also included. The τ value can be used to reduce false positives. In this work, τ and Φ were set to 50 and 6, respectively. Poor-quality channels were defined as channels with $PRD_{c}\geq th_{PRD}$.

#### 2.4.2. RMS Detection Method

The RMS detection method is based on the method described in [11]. This method is a rules-based method that identifies outliers based on thresholds of three features: (1) $P_{l/t}$, a measurement of low-frequency noise; (2) $P_{line/t}$, a measurement of power-line noise; and (3) RMS, a measurement of EMG amplitude. The following three features were calculated for each channel as follows:(4)$$P_{l/t}\left(n\right)=\frac{P_{0–12\mathrm{Hz}}}{P_{total}}$$
(5)$$P_{line/t}\left(n\right)=\frac{P_{50,100,150,200\mathrm{Hz}}}{P_{total}}$$
(6)$$RMS\left(n\right)=\sqrt{\frac{\sum_{i=1}^{N}x{\left(i\right)}^{2}}{N}}$$
where *n* is the channel number, $P_{l/t}$ is the relative power of low-frequency components, $P_{line/t}$ is the relative power of power-line interference components, $P_{0–12\mathrm{Hz}}$ is the sum of power densities from 0 to 12 Hz, $P_{50,100,150,200\mathrm{Hz}}$ is the sum of power densities of 50 Hz and its harmonics, $P_{total}$ is the sum of all power densities, RMS is the root mean square, x(i) is the recorded EMG signal, and *N* is the total number of samples.

Power ratio thresholds were calculated based on the interquartile range (*IQR*). First, a set of reference channels (ref) within 1.5×IQR was selected. Four threshold values were then determined as follows:(7)$$th_{l/t}=k_{1}(median\left(P_{l/t}\left(ref\right)\right)+1.5IQR\left(P_{l/t}\left(ref\right)\right)$$
(8)$$\begin{array}{c} th_{line/t}=k_{line}(median\left(P_{line/t}\left(ref\right)\right)+\\ 1.5IQR(P_{line/t}\left(ref\right)) \end{array}$$
(9)$$\begin{array}{c} th_{RMS_{low}}=min\left(\mu_{p_{a}},\mu_{p_{b}},\mu_{p_{c}}\right)-\\ k_{2}max\left(\sigma_{p_{a}},\sigma_{p_{b}},\sigma_{p_{c}}\right) \end{array}$$
(10)$$\begin{array}{c} th_{RMS_{high}}=min\left(\mu_{p_{a}},\mu_{p_{b}},\mu_{p_{c}}\right)+\\ k_{2}max\left(\sigma_{p_{a}},\sigma_{p_{b}},\sigma_{p_{c}}\right) \end{array}$$
where $k_{1}$, $k_{2}$, and $k_{line}$ are constants that are tuned to optimal values; μ and σ are the mean and standard deviation, respectively; and $p_{a}$, $p_{b}$, and $p_{c}$ are the sets of a channel and its closest two channels in the longitudinal direction ($p_{a}$) and each diagonal direction ($p_{b}$ and $p_{c}$), respectively. For this experiment, $k_{1}$, $k_{2}$, and $k_{line}$ were set to values of 3, 3, and 2.5 respectively.

Poor-quality channels were defined as any channels with $P_{l/t}\geq th_{l/t}$, $P_{line/t}\geq th_{line/t}$, $RMS\leq th_{RMS_{low}}$, or $RMS\geq th_{RMS_{high}}$.

#### 2.4.3. NMI Detection Method

The NMI detection method is based on the method described in [12]. The NMI feature indicates the mutual dependence of two EMG signals on a scale from 0 to 1, with 0 representing no dependence and 1 representing perfect correlation. The NMI between each channel and every other channel in the HD-EMG array was calculated [17]. This produced a 120 × 120 matrix (NMI), where NMI(i,j) is the *NMI* between the *i*th and *j*th channels. An “interaction” was defined as the number of times the *NMI* feature for a channel exceeded a predefined threshold. Channels with zero interactions are more likely to be of poor quality, as these electrodes are dissimilar to surrounding channels. Channels that achieved zero interactions with lower threshold values were considered poor-quality. A feature (*V*) was defined for each channel as the smallest threshold at which there were ≤5 interactions with other channels over the 10 s contraction. A threshold ($V_{th}$) for each array was calculated as follows:(11)$$V_{th}=median\left(V\right)-2std\left(V\right)$$

Channels with $V\leq V_{th}$ were labelled as poor-quality.

### 2.5. Reconstruction of Poor-Quality EMG Channels

In [13], various interpolation-based methods were explored for the reconstruction of a target channel in EMG arrays. The target channel is defined as the channel being interpolated. A variety of channel configurations were explored (nearest 4, 8, 12, and 24 channels (Figure 2)). The channel configuration refers to the selection of channels relative to the target channel used to perform the interpolation. These interpolation-based methods were evaluated by comparing the PRD and the Pearson correlation coefficient between a target channel (simulated as a missing channel to be reconstructed) and an interpolation of nearby electrodes.

The following two-dimensional (2D) interpolation techniques were explored:**Linear interpolation:** The target channel is estimated with linear interpolation over two dimensions (i.e., bilinear interpolation) [18];**Cubic interpolation:** A cubic polynomial is fit on each edge of a Delaunay triangulation [18];**Biharmonic spline interpolationL:** The target channel is estimated by determining the minimum curvature between irregularly spaced data in multiple dimensions [19]. In 2D, this process is equivalent to bicubic spline interpolation [20];**Nearest-neighbour interpolation:** The target channel is estimated as equivalent to the closest channel perpendicular to the muscle fibres.

Table 1 shows a summary of the 2D interpolation results [13]. A 2D biharmonic spline interpolation using the nearest 24 channels to the missing channel provided the best interpolation result (mean PRD = 15.5%, mean correlation = 0.98), and performance remained high even with fewer channels.

Then, 2D spline interpolations using the nearest 24 channels were applied to reconstruct poor-quality channels. For situations in which there were contaminated neighbouring channels or unavailable channels (e.g., the target channel was near the array border), the greatest number of possible clean channels was used to perform the interpolation. In the worst-case scenario, the non-contaminated channel identified nearest to the target electrode was used in place of the target electrode (nearest-neighbour interpolation). In this paper, we assume that when performing channel reconstructions, the vast majority of channels within the EMG array are of adequate quality. If many channels are noisy, the probability of a false negative may be increased (i.e., undetected poor-quality channel); however, the interpolation-based detection method detects many of the poor-quality channels, and with a large number of poor-quality channels detected, it may be more prudent to discard the EMG recording and rerecord the data than attempt to reconstruct these channels.

### 2.6. Evaluation

#### 2.6.1. Simulated Data

Each detection method was evaluated for each subject, SNR, and number of noisy channels, and the number of true positives, false positives, and false negatives for each subject was recorded. A true positive was defined as a single correctly identified noisy channel, a false positive defined as a single clean channel falsely identified as noisy, and a false negative was defined as a single noisy channel falsely identified as clean. These values were used to determine overall precision, recall, and F1 scores [21] across all subjects for each SNR level and number of noisy channels.

#### 2.6.2. Real Data

Each detection method was evaluated on a total of 24 EMG arrays (12 participants × 2 muscle groups). The precision, recall, and F -score for each method were identified. For real data, the ground truth was based on qualitative observations of “poor” or not “poor” (i.e., “adequate”, “good”, or “excellent”) by human raters. Quantitative evaluation of the reconstruction was not possible.

## 3. Results

### 3.1. Detection Results: Simulation

Table 2 shows the mean precision, recall, and F1 scores for one contaminated electrode for SNRs ranging from −20 to 15 dB. Table 3, Table 4 and Table 5 show the results for two, four, and eight contiguous noisy channels, respectively. The results for two, four, and eight random noisy channels are summarized in Table 6, Table 7 and Table 8, respectively. When the SNR is high, there may be no poor-quality channels detected, so the precision (and F1 score) is not a number. In the tables, “–” indicates these cases.

The F1 score for the interpolation-based detection method was ≥98.8% for SNRs ≤ 0 dB for all simulated results. F1 scores for the RMS method were ≥88.0% for SNRs ≤ −10 dB across all simulations. For the NMI method, for simulations with SNRs of 0 dB or lower, the F1 score was as low as 72.3%.

For a single noisy channel, the F1 score of the interpolation-based detection method remained above 99% for SNRs ≤ 0 dB. The F1 score of the RMS method was at least 98% for SNR ≤ −10 dB. For the NMI method, the F1 score was 81.5% for SNR ≤ −5 dB, with a precision of only 68.8%.

### 3.2. Detection Results: Real Data

#### 3.2.1. Rater Agreement

The kappa score of agreement between the human raters for poor quality vs. adequate or better quality was 0.85, indicating strong agreement [22].

#### 3.2.2. Detection

Table 9 summarizes the precision, recall, and F1 score for each detection algorithm for identifying poor-quality channels in the set of real HD-EMG data. The interpolation-based detection method provided the highest overall F1 score of 94.7%.

### 3.3. Reconstruction Results

Figure 3, Figure 4 and Figure 5 show examples of the process of detection and reconstruction. Figure 3 shows the RMS heat map of a representative HD-EMG array. A poor-quality channel, as indicated with red borders, was identified using the interpolation-based detection method. This poor-quality channel and its eight neighbouring channels are shown in the time domain in Figure 4. The poor-quality channel is illustrated in black. EMG signals for the eight adjacent channels are shown in blue. The poor-quality channel was interpolated with a 2D spline interpolation based on the eight nearest electrodes to produce a reconstructed result, as illustrated in yellow. Figure 5 shows the RMS heat map following the reconstruction of the poor-quality channel. As noted in Section 2.5, 2D spline interpolation was found to have a mean PRD of 15.5 ± 12.1% when using 24 channels to interpolate a known target channel. Figure 6 shows representative examples of target channels and their interpolations, with varying PRD values.

## 4. Discussion

### 4.1. Detection: Simulated Data

The three detection methods were tested using simulated poor-quality channels to study the effects of SNR and the number of noisy channels on efficacy.

#### 4.1.1. SNR of Noisy Channels

The NMI method had the lowest overall performance (F1 score = 81.5%) for SNRs ranging from −20 to 0 dB, and precision for this method was low (68.8%). The NMI performance degraded rapidly for SNRs greater than 0 dB. The RMS and interpolation-based detection methods were highly effective for identifying single noisy channels in the range of −20 to −10 dB (F1 scores of 98–100%). The recalls of these methods dropped sharply as the SNR of the contaminants increased; the RMS method degraded rapidly for SNRs over −5 dB, while the interpolation-based detection method had a better performance, degrading for SNRs greater than 0 dB.

In practice, weakly contaminated channels (i.e., SNR = 0–20 dB) are likely to be labelled as of adequate quality rather than as poor-quality channels. Thus, it is expected that recall would decline as SNR increases. Simultaneously, maintaining high precision is important, as false positives result in incorrect identification of adequate channels as poor-quality channels, causing unnecessary disruptions to the measurement process.

#### 4.1.2. Number of Noisy Channels

The performances of the RMS and interpolation-based detection methods declined as the number of noisy channels in the HD-EMG array increased, as expected. Additional noisy channels lower the thresholds used to identify outliers based on the PRD and RMS features, making noisy channels more difficult to detect. While performances declined, the interpolation-based detection performance remained high. For instance, the interpolation-based detection method had a precision ≥99.9% and recall ≥97.6% for eight noisy channels at 0 dB. The RMS method had 100% precision but very poor recall for multiple noisy channels, with recall ≥55.1% at −5 dB and ≥6.8% at 0 dB for eight noisy channels.

Counterintuitively, the precision of the NMI detection method improved across most SNR levels as the number of noisy channels increased up to four noisy channels, although precision was relatively low (precision ≥ 83.3 for SNR ≤ 0 dB). The threshold level used to identify noisy channels was calculated based on the median and standard deviation of the number of channel interactions. The ideal threshold was dependent on both the SNR and the number of noisy channels, and there was not a linear relationship between the median and standard deviation. The NMI method could be improved if adjustments were made to the threshold based on the estimated SNR level and the number of potential noisy channels.

#### 4.1.3. Location of Noisy Channels

The NMI detection method had similar performances for both contiguous and distributed noisy channels. This was expected, as the position of channels in the array was not accounted for in the NMI method. The interpolation-based and RMS detection methods were expected to perform better if noisy channels were dispersed throughout the array and not contiguous because these methods based their the features of these methods are on the relationship of channels within a local neighbourhood. However, for the interpolation-based detection method, there was minimal difference between random and contiguous groups of up to eight noisy channels, and the RMS method had minimal differences up to four noisy channels. However, for the RMS method, contiguous channels performed better across all SNR levels for eight noisy channels. For instance, at −20 dB, the RMS method provided a lower recall (89.9%) for randomly dispersed noisy channels than for contiguous channels (96.9%).

These simulations should be generalized to real HD-EMG with reservation. Only one type of contamination (random noise unique to each contaminated channel) was explored. In experimental situations, multiple sources of noise (e.g., power-line noise and motion artifacts) could contaminate the data, and the same noise source could contaminate multiple channels. Furthermore, the simulations explored additive noise, when in real HD-EMG data, poor-quality channels may manifest as low-amplitude recordings (i.e., due to electrode disconnection).

### 4.2. Real Data

All three tested detection methods were able to identify poor-quality channels to a moderate degree (Table 9). The RMS-based method had moderate performance (F1 score = 64.5%). The RMS method was good at identifying channels with anomalous RMS values within the electrode array (Figure 3). However, the RMS method was unable to identify poor-quality channels that had similar amplitudes (and, therefore, similar RMS feature values) but had significant differences in the time domain compared to nearby channels, which is indicative of contamination (Figure 7).

The NMI method was able to identify differences in the RMS feature (Figure 3), as well as differences in the time domain, even when the amplitude of the channels was around the same level (Figure 7). However, the NMI method had very poor precision and falsely identified many adequate channels as poor-quality.

The interpolation-based detection method provided the best overall detection performance for the dataset (F1 score of 94.7%). Based on further observation of the $PRD_{c}$ metric across all channels within each array, the parameters for the thresholds ($th_{PRD}$) were adjusted to τ=50 and Φ=3, increasing recall to 100%, with an updated F1 score of 96.4%.

Each detection method has the potential for improvement. Further refinement of the thresholds for all of these algorithms based on the dataset could improve recall. The NMI method has the longest computation time due to the requirement to calculate the NMI between each HD-EMG channel, which would make it unsuitable for real-time applications. Reducing the number of NMI calculations to a smaller neighbourhood around each channel would improve the computation time. The NMI method is also a global detection method, identifying outlier channels within the entire HD-EMG array, when it may be more appropriate to consider channels within a local context, as the EMG activity varies throughout the array. The interpolation-based detection method, which only considers four channels adjacent to the target channel, can be considered a complementary method to the NMI method in that it only considers a local context. The interpolation-based detection method was not tested for correctly classifying the border channels of the array. This method can be easily adapted for border channels (i.e., $PRD_{c}$ would be the composite feature of two PRDs for corner channels and three PRDs for all other border channels). This method may perform worse for border channels because fewer neighbouring channels would be available to calculate the PRD.

These results are limited by the relatively low number of poor-quality channels available in the real dataset (19 in the dataset). The distribution of poor-quality channels tended to involve a singular poor-quality channel or two contiguous poor-quality channels; therefore, the ability of each algorithm to identify clusters of four or more poor channels in real data could be not be evaluated. The ground truths for poor-quality channels were ratings performed by only two human raters. Additional raters could improve the reliability of the human ratings, although agreement between the two raters was high (kappa = 0.85, strong agreement). Finally, the suggested values for parameters τ and Φ were empirically determined and could be further optimized and tested on other datasets to ensure generalizability.

### 4.3. Reconstruction

Although it is not possible to empirically verify the interpolation results for the reconstruction of missing data, the reconstruction appears to be reasonable within the context of the HD-EMG array (Figure 4 and Figure 5). Based on simulations for reconstructing a known target channel (Table 1), the interpolation-based method is anticipated to provide good reconstructions.

## 5. Conclusions

Herein, we describe a novel interpolation-based process for the detection and reconstruction of poor-quality channels in HD-EMG arrays. Interpolation-based detection of poor-quality channels was effective in simulations for noisy channels up to 0 dB (F1 score ≥ 98.8%) and provided on-par or better results for all SNR levels and numbers of poor-quality channels when compared with RMS and NMI-based detection methods. Interpolation-based detection was also the best method for detecting poor-quality channels in real data (F1 score up to 97.4%). The detected poor-quality channels were successfully reconstructed via 2D spline interpolation.

There were several limitations of this research. The simulated poor-quality channels used in this study were developed with additive white Gaussian noise. The simulations could be expanded to study other types of common EMG contaminants (e.g., power-line interference and motion artifacts). The real data were limited by the low number of poor-quality channels. Finally, the proposed method was only tested on one dataset for bicep and tricep contractions at 30% MVC. This work could be extended to other datasets for different muscles and contraction conditions.

Future work will involve further tuning of detection parameters to improve recall of poor-quality channels, extending detection and reconstruction to a variety of HD-EMG datasets and exploring interpolation-based methods to detect and reconstruct poor-quality channels on the border of the HD-EMG array. The interpolation-based, RMS, and NMI detection methods may have some complementary features, combining methods to improve the overall system performance. The efficacy of these detection methods for a variety of specific contaminants (e.g., power-line noise and motion artifacts) will also be explored.

## Figures and Tables

**Figure 1 sensors-23-04759-f001:**
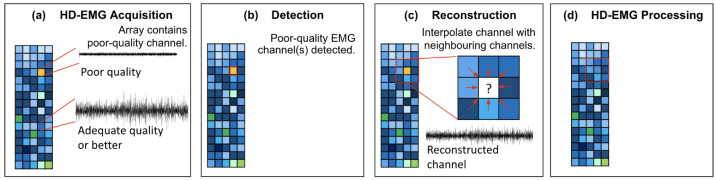
Block diagram of the proposed detection and interpolation process. (**a**) HD-EMG signals are acquired. (**b**) Poor-quality channel(s) are detected. (**c**) Neighbouring channels are used to reconstruct the detected poor-quality channels. (**d**) An HD-EMG array with the original dimensions preserved is available for further analysis.

**Figure 2 sensors-23-04759-f002:**
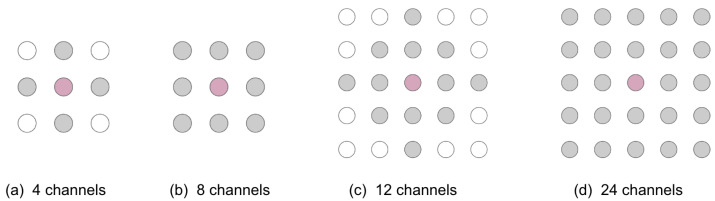
Channel configurations for interpolation of the nearest (**a**) 4, (**b**) 8, (**c**) 12, and (**d**) 24 channels. The target channel is indicated in pink. The channels used to perform the interpolation are indicated in grey.

**Figure 3 sensors-23-04759-f003:**
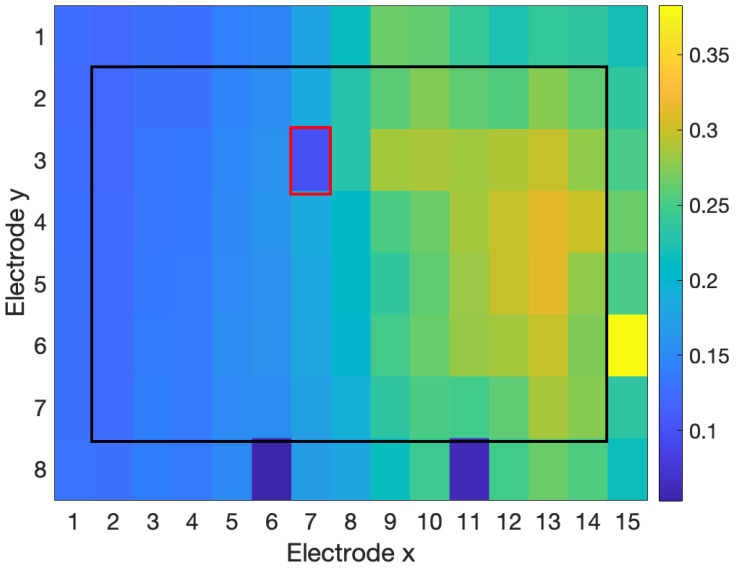
RMS heat map for outlier detection in real data. The interpolation-based detection method identified the poor-quality channel (indicated by red borders) with precision and recall of 100%.

**Figure 4 sensors-23-04759-f004:**
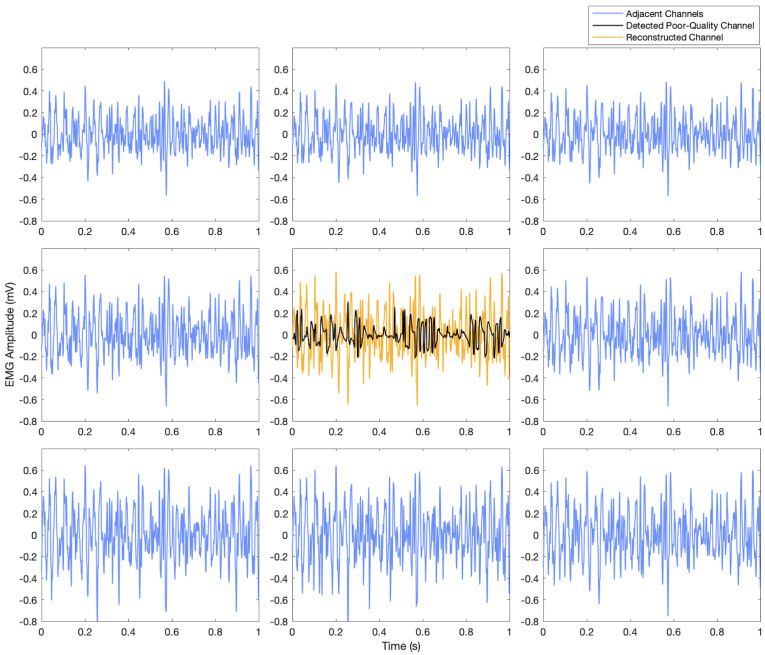
Detection and reconstruction of the poor-quality channel detected in Figure 3 using interpolation techniques. The channel identified by human raters as poor-quality is indicated in black, and the interpolated reconstruction of the channel is shown in yellow. The eight adjacent channels are illustrated in blue.

**Figure 5 sensors-23-04759-f005:**
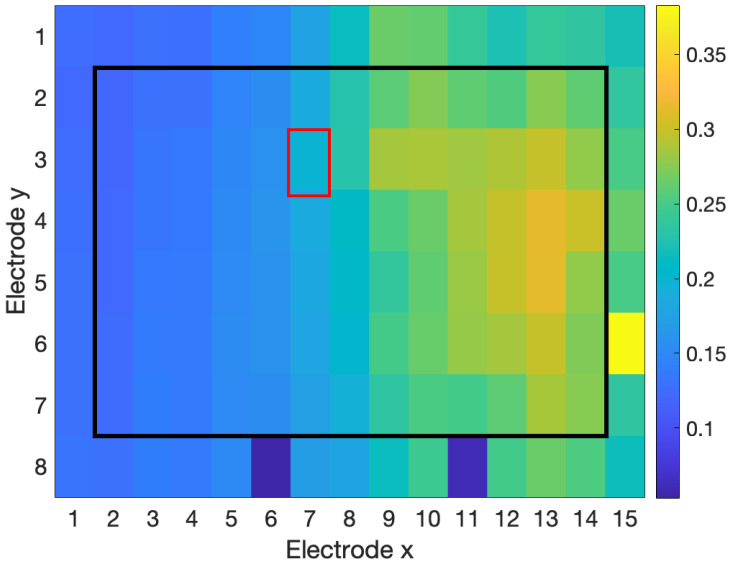
RMS heat map for the reconstruction of the poor-quality channel detected in Figure 3. The reconstructed channel is indicated with a red border.

**Figure 6 sensors-23-04759-f006:**
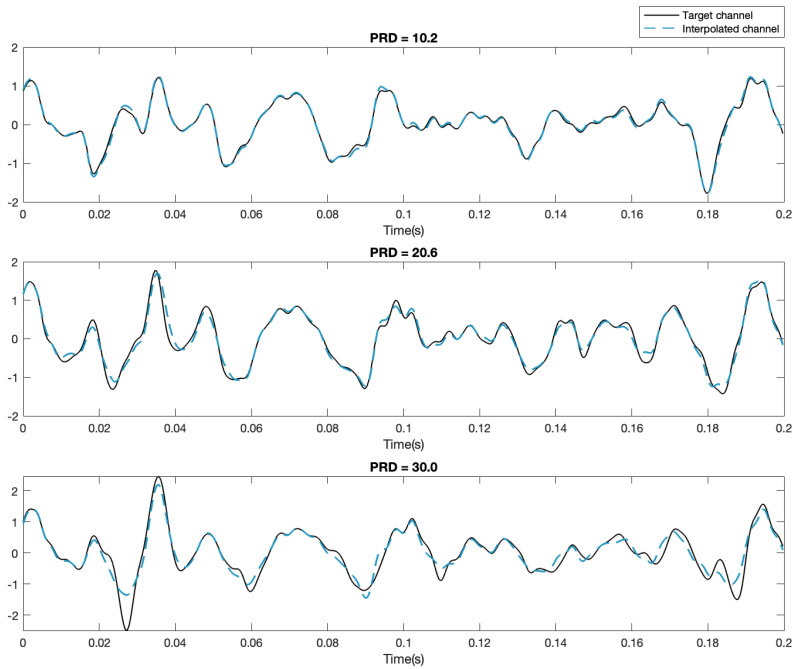
Sample target and interpolated HD-EMG channels for PRD = 10.2, 20.6, and 30.0%.

**Figure 7 sensors-23-04759-f007:**
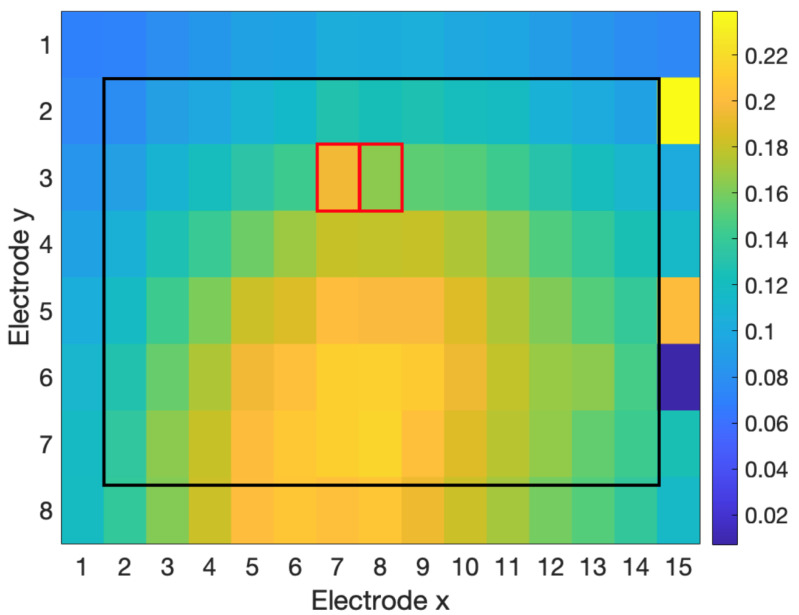
RMS heat map for outlier detection in real data. Poor channels identified by human raters are indicated in red. The RMS method was unable to identify either channel. The NMI and interpolation-based detection methods were able to detect the left outlier channel. The interpolation-based detection method detected both outlier channels after parameters were further tuned.

**Table 1 sensors-23-04759-t001:** Mean PRD for the reconstruction of target HD-EMG channels via interpolation. A lower PRD indicates that the reconstructed channel is more similar to the target channel.

Interpolation Method	4 Channels	8 Channels	12 Channels	24 Channels
Linear	21.6 ± 15.6	21.7 ± 15.7	21.9 ± 15.8	21.9 ± 16.0
Triangular Cubic	21.6 ± 15.6	19.8 ± 14.9	19.2 ± 15.2	19.3 ± 15.3
Biharmonic Spline	16.9 ± 12.1	17.0 ± 12.4	17.9 ± 10.8	15.5 ± 12.1
Nearest Neighbour	28.4 ± 31.9	28.4 ± 14.1	28.5 ± 14.2	28.5 ± 14.6

**Table 2 sensors-23-04759-t002:** Precision (P), recall (R), and F1 score (F) for one simulated noisy channel. “–” is indicates cases in which the precision and F1 score are not a number.

SNR (db)	Interpolation			RMS			NMI		
	P	R	F	P	R	F	P	R	F
−20	100	100	100	100	100	100	68.8	100	81.5
−15	100	100	100	100	100	100	68.8	100	81.5
−10	100	100	100	100	96.1	98.0	68.8	100	81.5
−5	100	100	100	100	71.3	83.3	68.8	100	81.5
0	100	98.2	99.1	100	24.8	39.7	61.1	100	75.9
5	100	19.1	32.1	100	0.30	0.60	41.7	45.5	43.5
10	–	0	–	–	0	–	25.0	27.3	26.1
15	–	0	–	–	0	–	10.0	9.1	9.5

**Table 3 sensors-23-04759-t003:** Precision (P), recall (R), and F1 score (F) for two noisy, contiguous channels. “–” indicates cases in which the precision and F1 score are not a number.

SNR (db)	Interpolation			RMS			NMI		
	P	R	F	P	R	F	P	R	F
−20	100	100	100	100	100	100	84.6	100	91.7
−15	100	100	100	100	100	100	84.6	100	91.7
−10	100	100	100	100	95.7	97.8	88.0	100	93.6
−5	100	100	100	100	68.9	81.6	88.0	100	93.6
0	100	97.8	98.9	100	22.8	37.1	84.6	100	91.7
5	100	6.0	11.3	100	0.30	0.59	66.7	45.5	54.1
10	100	0.45	0.89	–	0	–	41.7	22.7	29.4
15	–	0	–	–	0	–	18.2	9.1	12.1

**Table 4 sensors-23-04759-t004:** Precision (P), recall (R), and F1 score (F) for four noisy, contiguous channels. “–” indicate cases in which the precision and F1score are not a number.

SNR (db)	Interpolation			RMS			NMI		
	P	R	F	P	R	F	P	R	F
−20	100	100	100	100	99.8	99.9	93.0	90.9	92.0
−15	100	100	100	100	99.8	99.9	95.7	100	97.8
−10	100	100	100	100	96.2	98.1	95.7	100	97.8
−5	100	100	100	100	67.6	80.6	95.7	100	97.8
0	100	99.0	99.5	100	15.6	27.0	95.3	93.2	94.3
5	100	13.5	24.0	100	0.30	0.59	88.0	50.0	63.8
10	100	0.52	1.0	–	0	–	75.0	27.3	40.0
15	–	0	–	–	0	–	41.7	11.4	17.9

**Table 5 sensors-23-04759-t005:** Precision (P), recall (R), and F1 score (F) for eight noisy, contiguous channels. “–” indicates cases in which the precision and F1 score are not a number.

SNR (db)	Interpolation			RMS			NMI		
	P	R	F	P	R	F	P	R	F
−20	100	100	100	100	96.9	98.4	97.0	73.9	83.9
−15	100	100	100	100	95.1	97.5	98.8	93.2	95.9
−10	100	100	100	100	87.1	93.1	98.9	100	99.4
−5	100	100	100	100	55.7	71.6	98.9	100	99.4
0	100	99.9	99.9	100	6.8	12.7	98.7	88.6	93.4
5	100	19.8	33.1	100	0.22	0.44	96.1	55.7	70.5
10	100	0.33	0.66	–	0	–	93.8	34.1	50.0
15	–	0	–	–	0	–	70.0	15.9	24.9

**Table 6 sensors-23-04759-t006:** Precision (P), recall (R), and F1 score (F) for two noisy, random channels. “–” indicates cases in which the precision and F1 score are not a number.

SNR (db)	Interpolation			RMS			NMI		
	P	R	F	P	R	F	P	R	F
−20	100	100	100	100	99.9	99.9	83.3	90.0	87.0
−15	100	100	100	100	100	100	84.0	100	91.3
−10	100	100	100	100	94.4	97.1	88.5	100	93.9
−5	100	100	100	100	69.4	82.0	85.2	100	92.0
0	100	98.7	99.3	100	19.4	32.4	85.2	100	92.0
5	100	3.5	6.7	100	0.14	0.29	80.0	72.7	76.2
10	100	0.15	0.30	–	0	–	53.8	31.8	40.0
15	–	0	–	–	0	–	27.3	12.5	17.1

**Table 7 sensors-23-04759-t007:** Precision (P), recall (R), and F1 score (F) for four noisy, random channels. “–” indicates cases in which the precision and F1 score are not a number.

SNR (db)	Interpolation			RMS			NMI		
	P	R	F	P	R	F	P	R	F
−20	100	100	100	100	98.8	99.4	87.9	64.4	74.4
−15	99.9	100	100	100	97.5	98.7	93.0	85.1	88.9
−10	99.9	100	100	100	89.3	94.4	92.6	100	96.7
−5	99.9	100	99.9	100	63.5	77.6	93.8	100	96.7
0	100	98.0	99.0	100	18.3	31.0	94.0	100	96.9
5	100	3.8	7.4	100	0.37	0.73	90.6	63.0	74.4
10	100	0.07	0.14	–	0	–	75.0	32.6	45.5
15	–	0	–	–	0	–	41.7	10.4	15.7

**Table 8 sensors-23-04759-t008:** Precision (P), recall (R), and F1 score (F) for eight noisy, random channels. “–” indicates cases in which the precision and F1 score are not a number.

SNR (db)	Interpolation			RMS			NMI		
	P	R	F	P	R	F	P	R	F
−20	99.9	100	100	100	89.9	94.7	98.1	57.3	72.3
−15	99.7	100	99.9	100	87.0	93.0	97.4	82.4	80.3
−10	99.9	100	99.9	100	78.6	88.0	97.9	100	98.9
−5	99.9	100	99.9	100	55.1	71.0	98.9	100	99.4
0	99.9	97.6	98.8	100	13.3	23.4	97.6	89.9	92.6
5	98.3	2.2	4.2	100	0.18	0.37	96.0	52.2	67.6
10	100	0.26	0.51	–	0	–	92.9	28.0	43.0
15	–	0	–	–	0	–	71.4	10.8	18.7

**Table 9 sensors-23-04759-t009:** Precision (P), recall (R), and F1 score (F) for detection of poor-quality electrodes in the real dataset.

Interpolation			RMS			NMI		
**P**	**R**	**F**	**P**	**R**	**F**	**P**	**R**	**F**
94.7	94.7	94.7	83.3	52.6	64.5	35.6	84.2	50.0

## Data Availability

Not applicable.

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
