# Peer review of "Detection and Reconstruction of Poor-Quality Channels in High-Density EMG Array Measurements"

_sensors, 2023, doi:10.3390/s23104759_

Round 1
Reviewer 1 Report
First of all: congratulations to a good job - interesting topic generalizable to a lot of other scientific fields.
Comments and questions:
r. 93: develop = developed? or = are developing?
r. 101: ... 30%... How did you know that?
r. 140: ... poor quality channels... why did you exclude them?
Chap. 2.5: closer explanation/description and figures and graphs will be helpful
Table 1: triangular...
Suggestions for better understanding of the text by the reader:
It is not clear to me how the target channels are defined/created.
Are you sure that you really do reconstruction? Is here a risk that you eliminates differences only?
For both: maybe a closer justification helps...
Finally: I feel a big potential for further the following work and its eventual practicable impact. Thus I believe that your article is interesting already in its current form and all my comments and questions, understand like proposals how to make your text more reader-friendly please.
Reviewer 2 Report
The manuscript presents a novel method for detection and reconstruction of low-quality channels in EMG measurements based on 2D interpolation, and compare it with two other established methods, with very good results. The manuscript is well-written, with almost no English issues, and the results are sound. I have only a few suggestions to improve the quality of the manuscript, mainly related to the detailing of the methods, as follows:
- throughout the manuscript, units shall be separated from values (e.g. "99.9 %" instead of "99.9%"; "10 dB" instead of "10dB", etc)
- the thousands separator shall not be used, and it can be replaced by a space of by nothing: "20 160" or "20160" instead of "20,160";
- in section 2.4.1, which interpolation method was used for the detection? That is, how the target channel is interpolated from the neighboring channels?
- in section 2.4.3, how is the NMI calculated? Please provide equations.
- in section 2.5, please provice equations for each type of interpolation;
- in section 3.1, please clarify if, when there is more than one noisy channel, the objective is to detect ALL noisy channels (and thus if a true positive is computed only in this case);
- section 3.2 should start only after Table 8;
- in section 3.2.1, please provide a reference (and/or equations) for the kappa score;
- in figure 3, it is (quite) difficult to distinguish between the original signal and the interpolated signal;
Reviewer 3 Report
This paper proposes an interpolation-based method for the detection and reconstruction of poor-quality channels in HD-EMG arrays. The proposed detection method identified artificially contaminated channels of HD-EMG for signal-to-noise ratio (SNR) levels 0 dB and lower with ≥ 99.9% precision and 6
≥ 97.6% recall. Generally, the topic is very interesting. However, the novelty needs to be higlightes. Moreover,several comments need to be addressed before accepting this paper
In the abstract could you explain the novelty of the proposed method compared to existing methods for detection and reconstruction of poor quality channels?
EMG should be mentioned in full in the abstract.
Introduction:
Could you please summarize related studies in a table and mention the limitations of related studies?
Could you please highlight novelty and contributions?
Also, could you please add a paragraph describing the organization of the paper by the end of the introduction section?
Methods:
Please explain briefly different blocks of Figure 1
Could you add EMG samples of the dataset?
Please add more details regarding particpants information, inclusion/exclusion criteria.
Why only two raters rated the quality of the EMG signals?
Please justify the use of features.
Why did not the authors used ant methods for filtering and denoising?
Experimental Results
Please define the performance metrics that you employed.
Also add accuracy and confusion matrices.
Why did not the authors compared their results with previous studies based on the same dataset.it is very important to do this comparison.
Please add your limitations
Round 2
Reviewer 3 Report
I would like to thank the authors for properly addressing my comments.